## Research Article

self-harm; suicide; interventions; counselling; low- and middle-income

**Corresponding author:**
Shilpa Aggarwal;
Email: shilpazq@gmail.com

# Effectiveness of ATMAN psychological intervention in reducing self-harm in young people in India: a mixed method case series

Shilpa Aggarwal[1,2,3] 🔾, Michael Berk[1], Nilesh Shah[4], Anokhi Shah[5], Dimple Kondal[6], George Patton[3,7] and Vikram Patel[8] 🔾

[1]IMPACT – The Institute for Mental and Physical Health and Clinical Translation, School of Medicine, Barwon Health, Deakin University, Geelong, Australia; [2]Child and Youth Mental Health, Mental Health Speciality Services, Gold Coast, Australia; [3]Centre for Adolescent Health, Murdoch Children's Research Institute, Melbourne, Australia; [4]Lokmanya Tilak Municipal Medical College & General Hospital Municipal Corporation, Mumbai, India; [5]Public Health Foundation of India, New Delhi, India; [6]Centre for Chronic Disease Control, New Delhi, India; [7]Department of Paediatrics, University of Melbourne, Melbourne, VIC, Australia and [8]Global Health and Social Medicine, Harvard T H Chan School of Public Health, Boston, MA, USA

## Abstract

There is a scarcity of psychological interventions for self-harm in young people, either developed or adapted for use in low and middle-income countries (LMICs). ATMAN *is* a psychological intervention developed in India for youth with three key modules: problem-solving, emotion regulation and social network strengthening skills in addition to crisis management. ATMAN was delivered in 27 youth with a history of self-harm (14–24 years old) sequentially by a specialist and it a non-specialist counsellor. Out of 27, 18 youth who started the ATMAN intervention completed it, and 13 completed the 10-month follow-up. There was a significant reduction in post-intervention scores on Beck's Scale for Suicidal Ideation (BSI) (mean difference [confidence interval]: 14.1 [17.2, 10.9]) and Patient Health Questionnaire (PHQ-9) (9.6 [12.8, 6.4]) from the baseline scores, irrespective of who delivered the intervention (non-specialist vs. specialist). The difference remained significant at the 10-month follow-up (BSI: 17.0 [20.5, 13.6] and PHQ-9: 10.5 [14.5, 6.6]). Themes such as improved understanding of self-harm acting as a deterrent, using ATMAN strategies to deal with daily life distress, and the importance of addressing stigma in self-harm emerged during the qualitative interviews. Although requiring further evaluation, ATMAN shows promise as a scalable intervention that can be used in LMICs to reduce the burden of suicide in young people.

## Impact Statement

Seventy-eight percent of global suicides occur in low- and middle-income countries (LMICs). Self-harm, which is defined as any act of intentionally causing harm to oneself, such as self-cutting or ingesting a toxic substance irrespective of the type, motive or intent, is an identified antecedent of suicide. The World Health Organisation recommends using psychological interventions for self-harm to reduce the burden of suicide. However, the psychological interventions with the most evidence for reducing self-harm, such as dialectical behaviour therapy for adolescents (DBT-A), have been developed and tested in high-income countries (HICs) without being adapted for the use in youth in LMICs, where 90% of 1.2 billion adolescents live. Furthermore, many of these psychological interventions are resource intensive, which makes them difficult to use in low-resource settings.

ATMAN is one of the very few psychological interventions developed with inputs from expert service providers as well as service users with lived experience of self-harm, and integrating it with the available scientific evidence, to suit the needs of youth who self-harm in India and other limited resource settings. Consisting of three key and three optional modules, ATMAN shares a parsimonious set of common elements with the current psychological treatments available for youth self-harm that have been developed and tested in HICs. In the current study, we show that ATMAN intervention, when delivered by specialist and non-specialist counsellors alike, helps in reducing self-harm thoughts and behaviours in young people who self-harm. The positive effects were maintained at a 10-month follow-up. Our results are significant in the absence of scalable interventions available for youth self-harm that can be used in LMICs and limited evidence for the effectiveness of available resource-intensive interventions, such as DBT-A, in reducing self-harm in young people. Interventions that show effectiveness in varied cultural contexts can help us develop more effective interventions. Furthermore, this will guide the service providers and the funding agencies about the optimal forms of psychological interventions to invest in.

## Background

Self-harm, defined as any act of intentionally causing harm to oneself, such as self-cutting or ingesting a toxic substance, irrespective of the type, motive or intent, is an identified antecedent of suicide (Hawton et al. 2012; Moran et al. 2012). About 50–85% of persons who self-harm attempt suicide at least once during their lifetime (Hawton et al. 2015; Whitlock et al. 2013). In the year following self-harm, 10- to 18-year-olds have 30 times higher risk of suicide as compared to the general population (Hawton et al. 2020). Psychosocial treatments for self-harm are among the key recommendations of the World Health Organisation (WHO) to reduce the burden of suicide (World Health Organization 2014). Furthermore, the available guidelines recommend using psychological interventions that are effective and accessible for everyone who self-harm to prevent its recurrence (National Institute for Health and Care Excellence, N.I.C.E. 2022). However, the evidence regarding the effectiveness of available interventions in reducing self-harm in children and adolescents is inconclusive (Witt et al. 2021). Most psychological interventions that have shown a positive effect on youth self-harm, such as dialectical behaviour therapy for adolescents (DBT-A), were developed and tested in high-income countries (HICs) and have not been adapted for use in youth in low- and middle-income countries (LMICs) where 90% of 1.2 billion adolescents live (Aggarwal et al. 2021b; UNICEF & WHO 2022; Witt et al. 2021). A systematic review of psychosocial interventions for self-harm in LMICs found contact using postcards, cognitive behaviour therapy (CBT) and volitional help sheets to be effective in reducing suicidal ideations in LMICs (Aggarwal et al. 2021b). In addition, two psychotherapies have been culturally adapted for use in self-harm in adults in LMICs, that is, DBT adaptation in Nepal and an adaptation of problem-solving therapy in Pakistan (Aggarwal et al. 2021b; Knipe et al. 2019). Psychological interventions, either developed or adapted, for use in LMICs are likely to be more acceptable and useful for young people residing in such settings (Aggarwal et al. 2021b; Knipe et al. 2019). Psychological intervention adaptations should take into account the cultural knowledge about the condition, conceptualisation of the treatment and culturally attuned treatment methods and goals (Castro et al. 2010). Such adaptations, when used at a large scale, may help us to generate definitive evidence for the effectiveness of psychological interventions as well as identify the reasons for the limited effectiveness of the available psychosocial interventions. In addition, it will allow the pragmatic use of resources in HICs (Singla et al. 2017).

With extremely limited mental health workforce, using resource-intensive treatments for self-harm, such as DBT-A, is impractical to implement in routine healthcare settings in India (Aggarwal et al. 2024). One way for the health services in low-resource settings to provide psychosocial interventions to young people who self-harm is by using non-specialist providers following training, supervision and collaborative relationships with specialised providers, without formal advanced training or certification in mental health (Aggarwal et al. 2021a, Patel et al. 2023). This is to bolster the limited existing supports and not to replace the existing services. ATMAN is one such psychological intervention for self-harm in young people developed in India (Aggarwal et al. 2021a). "ATMAN" is a Sanskrit word describing "self" or "self-existent essence that functions in harmony with the Universe". ATMAN, with the potential to be delivered by counsellors and therapists with different levels of training and experience (including non-specialist providers), can be integrated at various levels of health services to ensure the availability of psychological intervention to youth who self-harm. The intervention consists of five to eight sessions and three key elements: problem-solving, emotion regulation and social network strengthening skills in addition to crisis planning. ATMAN is designed to be delivered by non-specialist providers. The intervention was delivered in a series of young people with self-harm, wherein a few modifications were introduced to improve upon its acceptability (Aggarwal et al. 2024). Culturally attuned case vignettes and strategies are part of the treatment schedule.

The current mixed method case series was conducted to generate the preliminary effects of ATMAN intervention in reducing self-harm thoughts when delivered by specialist and non-specialist providers; to assess if the effects were sustained at the 10-month follow-up; and to assess the intervention experience at each step. The study was based in Mumbai, the most populous city in India, with a population of about 22 million (UN World Urbanization Prospects).

## Methods

The current study used a mixed method case design to derive a quantitative estimate of the effects of the intervention and to complement this observation with a nuanced understanding of the subjective experiences of young people receiving the intervention (Johnson and Schoonenboom, 2016). The recruitment for the study occurred between January 2021 and November 2022 from an outpatient psychiatry department of a public sector tertiary hospital in Mumbai, India. Consecutive youth (14- to 24-year-olds reflecting mid to late adolescence, an age of higher prevalence for self-harm) who had self-harmed in the month before their presentation to the outpatient department were approached for recruitment (Patton et al. 2007). Informed consent (assent if the participant was under 18 years of age, with formal consent from parents/legal guardians) was obtained from all participants. Proficiency in written and spoken Hindi or English was mandatory to participate in the study procedures and treatment. Youth with conditions that could interfere in their capacity to participate in treatment, such as intellectual disability, acute psychosis or a medical condition, were excluded from the study.

## Intervention

ATMAN is a psychological intervention developed using a systematic, sequential approach. The steps included identifying prioritised outcomes for youth who self-harm in India with the help of lived experience consultants; selecting feasible and acceptable elements to achieve the outcomes from the distillation of self-harm interventions developed in HICs (those trialled and found to be effective in LMICs) intervention development workshops with mental health professionals and youth to finalise elements; and a review of relevant treatment manuals to decide on the treatment framework and to finalise the treatment structure and schedule. The details of the steps have previously been published along with the complete intervention design process(Aggarwal et al 2020; Aggarwal et al 2021a; Aggarwal et al. 2021b). The session-wise details of the intervention and the related resources (session scripts, outcome measures and handouts) have been published as an intervention manual (Aggarwal et al 2024).

The cultural adaptations in ATMAN intervention include the following:

**Intervention elements:** During the intervention development phase, elements well suited to the Indian cultural context were selected from the global evidence. The selection process was based on the findings of the intervention development workshops and in-depth interviews with lived experience consultants and mental health professionals.

**Module contents:** Furthermore, the explanatory models of young lived experience consultants informed various modules (e.g. formulation of self-harm) and explanations included in the intervention. The contents of the intervention were matched with the local attitudes (e.g. understanding of self-harm) and the available local supports (e.g. strengthening the social network module).

**Case vignettes:** The case vignettes and scripts included in the ATMAN intervention manual are culturally relevant (based on real-life scenarios from the formative phase) and focus on problems unique to young people in the Indian context.

The intervention consists of three key modules, three optional modules and a crisis management module (see Supplementary Table 1) delivered over five to eight sessions. ATMAN shares a parsimonious set of common elements with the current psychological treatments available for youth self-harm that have been developed and tested in HICs.

Each module consists of a meaningful unit to bring about a specific treatment outcome. The three key modules include problem-solving, emotion regulation and social network strengthening. In addition, three optional modules include an assertiveness skills training module, family module and a substance-use module. These optional modules can be used at the discretion of the counsellor and the participant, if deemed clinically appropriate.

Assertiveness skills session deals with information about the purpose that assertiveness skills serve in reducing distress for young people in difficult situations. The family session involves providing information to the family members about self-harm, identifying ways in which the family could support the youth who self-harm, promoting a positive communication style among family members and addressing any questions that family members may have about self-harm. In addition, young people had the option of including family members in any of the sessions related to the key modules. This was to allow young people to trust the process and ensure greater self-efficacy. Substance-use session, based on motivational interviewing techniques, involved making a connection between sel-harm and substance use (e.g., alcohol and cannabis). In addition, the counsellor helps the young person to identify the reasons for their ongoing substance use.

The initial assessment and the first session were conducted in-person for all the participants. Follow-up sessions occurred either in-person in the research office or online in a secure, private space. The sessions were individual, occurring every week. The overall duration of the intervention (number of days the participant remained engaged with the counsellor) varied according to the number of sessions delivered based on the needs of the young person and the availability of the young person. Participants could collaboratively work with the counsellor to end the intervention at any time after receiving the key modules.

Intervention was delivered by a specialist provider (child and adolescent psychiatrist, or a clinical psychologist) or by a non-specialist counsellor (with a bachelor's in psychology). Consecutive youth meeting the inclusion criteria were recruited and allocated sequentially to receive the intervention from either the specialist or the non-specialist counsellor.

In India, an accreditation as a specialist mental health provider requires an advanced course of a minimum of 2 years duration (e.g., M-Phil) following a master's degree in psychology (Sharan and Tripathi 2021). ATMAN can be delivered by non-specialist providers with some prior mental health training, which includes (but is not limited to) social work (and youth work), student well-being trained and supported teachers and nurses (Aggarwal et al. 2024). Training in ATMAN consists of reading and writing exercises, group discussions and seminars. Core training is conducted in a classroom setting over 30 h (spread over 3–4 days). In addition, counsellors are expected to observe at least one session of each ATMAN module while it is being conducted by a professional therapist as part of their training.

In the current study, the non-specialist counsellors had no experience in delivering therapy but knew the principles of CBT. The specialist providers (SA and AS) trained the non-specialist counsellors in the intervention structure, and the content of each session, before they started delivering the intervention. The specialist providers used video-recorded sections of their own intervention sessions and role-play to discuss various elements of each session. In addition, the video recordings of intervention sessions delivered by the counsellors were used to give feedback by the specialist providers during regular supervision. This helped in the adequate delivery of the intervention components.

### Outcome measures

A structured assessment was conducted at the start (baseline assessment before the treatment was commenced), end (scheduled to occur at 6–8 weeks from the start) and at a 10-month follow-up from the beginning of the treatment. In addition, mid-intervention progress monitoring (during the third or fourth intervention session) was used to assess whether the intervention was progressing in the right direction.

### Quantitative measures

We used **Beck's Scale for Suicidal Ideation (BSI)** to measure suicidal ideations at the baseline, end of intervention and follow during-up (Beck and Steer 1991). The BSI is a 19-item self-report measure, which is rated on a three-point scale from 0 to 2, to measure the intensity of an individual's attitudes, behaviours and specific thoughts about suicide within the week preceding the assessment. Higher scores on the scale indicate more severe suicidal thoughts and intent (a score below 6 is considered a non-suicidal score). The internal consistency coefficient for BSI has previously been reported as 0.96 and test–retest reliability as 0.88 (Pinninti et al. 2002). The BSI has been used in Indian settings and its inter-rater reliability in college students was 0.83 (Singh et al. 2012).

**The Columbia-Suicide Severity Rating Scale (C-SSRS)** recent screener was used to assess any suicidal thoughts and plans at the beginning of each session and during follow-up, as part of risk assessment. The C-SSRS has two initial screening questions, followed by an additional question for those with no suicidal thoughts or behaviour and three questions for those who do. The International Consortium for Health Outcomes Measurement recommends using C-SSRS for a quick assessment of suicidality in adolescents (as a screen) (Krause et al. 2021). It has been used in Indian and other linguistically diverse contexts and its clinician-reported version has shown sensitivity to change and good reliability in adolescents and adults (Kilincaslan et al. 2019; Chaudhary et al. 2016). The C-SSRS guided the formulation of risk and the corresponding level of clinical management. Affirmative responses restricted to Items 1 or

2 indicated low risk; Item 3 indicated moderate risk and Items 4, 5 or 6 indicated high risk (Bjureberg et al. 2022).

The **Patient Health Questionnaire (PHQ-9)** is a nine-item depression assessment scale developed from the evaluation of mental disorders in primary care (Kroenke et al. 2001). We administered the PHQ-9 at the start of the intervention, end of the intervention and during the follow-up assessment. It is validated for use in Indian adolescents. Each item on the PHQ-9 is scored from 0 ("not at all") to 3 ("nearly every day"). Mild depression is indicated by a score of 5–9, moderate depression by a score of 10–14, moderately severe depression corresponds to a score of 15–19 and a score of 20–27 is considered as severe depression (Ganguly et al. 2013).

**Functional Assessment of Self-Mutilation (FASM)** is a self-report tool that assesses the methods, frequency and functions of non-suicidal self-injury. It has shown adequate psychometric properties in both the general adolescent population and psychiatric cohorts and was used for a cross-sectional survey in India (Esposito et al. 2003; Guertin et al. 2001; Kharsati and Bhola 2015; Penn et al. 2003). We administered the FASM at the beginning and end of the intervention as well as during the follow-up. When administered at the end of the intervention, the timeframe for the FASM questions was from the start to the end of the intervention.

The **Global Assessment of Functioning Scale (GAS)** is a brief clinician-rated single-item measure of global functioning. It assesses the functioning of individuals during one year prior to administration and is widely used in India (Menon et al. 2013). The Children's GAS (CGAS) is adapted from adult GAS and developed for children and adolescents aged 4–16 years. It is also widely used in the Indian context (Srinath et al. 2005). CGAS is used by clinicians to score the functioning of children and adolescents on a scale from 1 to 100, from "extremely impaired" (1–10) to "doing very well" (91–100). The scale has demonstrated good inter-rater and test–retest reliability. In the current study, the scores on GAS (for participants 18 years and above) or CGAS (for participants below 18 years) were listed by the study clinician during every visit.

### Qualitative measures

The **Psychological Outcome Profiles (PSYCHLOPS)** is an individualised, client-generated outcome measure using open-ended questions based on the person's problems. The questionnaire lends itself to the problem-solving elements of the ATMAN intervention and helps in a therapeutic dialogue. We used teen and adult pre-therapy, mid-therapy and post-therapy versions in the current case series. The measure was administered at the start and end of the intervention for young people who received less than five sessions. For the rest of the participants, an additional administration occurred either during the third or fourth session to track the progress of the intervention. Two large randomised controlled trials have used PSYCHLOPS to test the effectiveness of PM+ (WHO intervention) in reducing symptoms of common mental disorders in Pakistan (similar sociocultural context to India) and Kenya (Dawson et al. 2015).

For the qualitative interviews, we developed an interview guide to explore the basic processes of recruitment and engagement, any difficulties in understanding the intervention elements, experiences of young people during the intervention and utility of treatment strategies (see Supplementary Table 5). Follow-up interviews focussed on the ongoing use of strategies, challenges young people may have faced during the follow-up period and reflections and recommendations for the ATMAN intervention to make it more

useful. Two authors (SA and AS) completed the exit and follow-up interviews for the participants who completed the intervention in a planned manner. SA interviewed participants who received the intervention from AS, while AS interviewed those who received the intervention from SA and the counsellors.

### Risk management

A risk management plan was developed as part of the study protocol. This plan used the responses of C-SSRS and BSI. The participants were informed about high-risk situations in which the research team was obliged to contact the family members/supports identified by them. These situations involved significant threats to the safety of the participants or others around them, due to their thoughts and/or actions.

Three ethics committees reviewed and approved the study: Institutional Ethics Committee Lokmanya Tilak Municipal Medical College and General Hospital, Mumbai-IEC/787121, Institutional Ethics Committee Public Health Foundation of India, Gurgaon, Haryana IEC-366.1/17 and Deakin University Human Research Ethics Committee, Melbourne Burwood Campus, Australia x-ref 2020-230.

### Statistical analysis

The pre-intervention, post-intervention and the 10-month follow-up scores of continuous variables (BSI and PHQ-9) were examined using a linear mixed-effects model to estimate the difference in mean over the study period (post-intervention and 10 months) from baseline. In the unadjusted analysis, the model included outcome (BSI and PHQ 9 scores) and time (baseline, post-intervention and 10 months) with participants treated as a random effect. Model 1 included the duration of the intervention to examine if it influenced the change in scores on the outcome measures. Model 2 further added age and sex to Model 1. Model 3 added the type of provider (specialist vs. non-specialist) to Model 2.

The predicted marginal estimates with mean and 95% confidence interval were reported. A $p$-value of less than 0.05 (two-sided) was considered statistically significant in all results. The mixed-effects model assumed the missing at random assumption.

A $\chi^2$-test was conducted to compare the baseline characteristics of participants who completed the intervention with those who dropped out (see Supplementary Table 2). Paired-sample $t$ tests were used to analyse the differences in pre- and post-intervention scores on the BSI and PHQ-9.

The values on the FASM were converted into categorical ratings of 0, 1, 2, where 0 indicates no self-harm, 1 represents one to five times and 2 denotes more than five times. This conversation aimed to provide an estimate of the frequency of non-suicide self-injury. The percentage of participants in each category of FASM was calculated in the year prior to the intervention and during the follow-up period.

We recorded qualitative interviews digitally, translated the interviews into English and transcribed them verbatim. Two authors (SA and AS) analysed the interviews using phenomenological thematic analysis (Vaismoradi et al. 2016). An inductive approach was used to understand the phenomena and identify the themes linked to the data, without trying to fit the data in a pre-existing coding framework. For analysis, each transcript was read multiple times and the transcripts were open coded for key themes to capture the essential qualities of the interview, including

examples and summaries. We decided on a final set of themes after resolving the differences by discussion.

## Results

### Sample

Out of 56 young people approached for the study, 40 consented to participate. Recruitment was based on the communication from the treating team. The intervention had to be discontinued in 2 young people after their 18 first session due to significant cognitive deficits in one and delusions in the other. In total, 27 young people (n-13 with specialist provider and n-14 with non-specialist provider) began the ATMAN intervention. 18 participants completed the intervention, including 4 participants in English and 14 in Hindi. A 10-month follow-up was completed for 13 young people (8 with a specialist provider and 5 with a non-specialist provider) (see Figure 1). The median duration of engagement during the intervention delivery was 40 days, with an average of five sessions, each lasting 50-min.

Post-intervention assessments occurred between 6 and 8 weeks for 17 young people who completed the intervention, with a single participant having the assessment at 5 weeks due to an out-of-area move. With the exception of 2 participants (with no social network-related difficulties), everyone received the key modules. The number of sessions received by the participants are shown in Figure 1.

Female participants constituted three-quarters of the sample with a greater percentage of participants in the 19–24 age range compared to those aged 14–18 (Table 1). Over half of the sample was diagnosed with either a depressive or anxiety disorder, while a quarter received a diagnosis of a personality disorder or traits (see Table 1). A comparison of baseline characteristics of treatment completers versus those who dropped out showed a significantly larger proportion of treatment completers to be on antidepressant medications and scoring higher on BSI when compared to young people who dropped out of intervention (see Supplementary Table 2).

### Outcomes

There was a significant reduction in the scores on the BSI and PHQ-9 post-intervention irrespective of who delivered it (non-specialist vs. specialist provider), age, gender and duration of the

**Table 1.** Characteristics of young people with self-harm who completed ATMAN intervention and 10-month follow-up

| Gender | Treatment initiation (n-27) | Treatment completion (n-18) | 10-month follow-up (n-13) |
|---|---|---|---|
| Female (%) | 18 (67%) | 13 (72%) | 10 (77%) |
| Age in years (%) | | | |
| 14–18 (%) | 11 (41%) | 7 (39%) | 5 (38%) |
| 19–24 (%) | 16 (59%) | 11 (61%) | 8 (62%) |
| Mental health diagnoses | | | |
| Serious mental illness (schizophrenia, bipolar disorder) | 5 (19%) | 3 (17%) | 2 (15%) |
| Depressive and/or anxiety disorders | 14 (52%) | 10 (56%) | 8 (62%) |
| Personality disorders/ traits | 7 (26%) | 5 (28%) | 3 (23%) |
| Substance use, neurodevelopmental disorders | 4 (15%) | 3 (17%) | 3 (23%) |
| Medications | | | |
| Antidepressants | 18 (67%) | 14 (78%) | 10 (77%) |
| Antipsychotics (oral/depot) | 9 (33%) | 5 (28%) | 1 (8%) |

intervention. Furthermore, these scores remained significantly lower at the 10-month follow-up compared to the baseline scores on both measures (see Tables 2 and 3, Figure 2). The paired-sample *t* tests showed a significant reduction in pre- and post-intervention scores for the BSI and PHQ-9 among the 18 young people who completed the intervention (see Supplementary Table 3).

A minor increase in PHQ-9 scores in two participants (1 point and 5 points at post-intervention, and 8 and 6 points at the 10-month follow-up) were associated with a simultaneous decrease in BSI scores (a decrease of 2 and 1 point, respectively, from baseline at post-intervention, and 8 and 9 points at the 10-month follow-up).

Supplementary Table 4 shows a comparison of the mean and median values of BSI and PHQ-9 at baseline, post-intervention and at the 10-month follow-up. This comparison includes participants

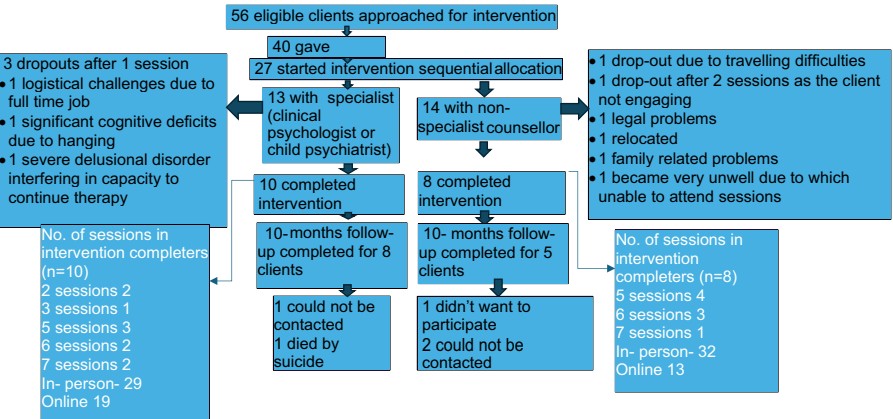

**Figure 1.** Participant selection flowchart.

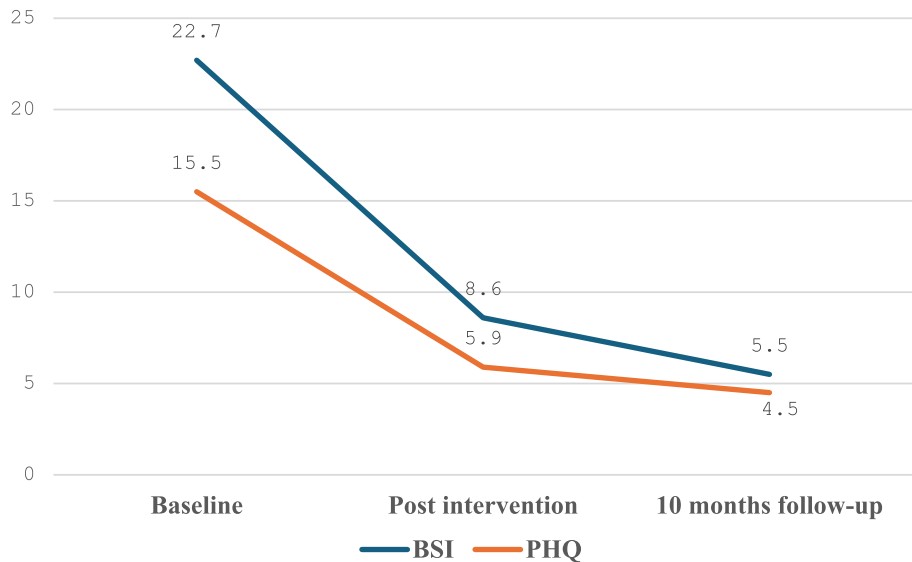

**Figure 2.** BSI and PHQ-9 Mean Scores of follow-up completers at baseline, intervention completion and 10-months follow-up.

receiving the intervention by specialist providers and those who received it from non-specialist providers. The values were comparable for both the outcome measures at all three time points.

During the baseline assessment, 3 out of 13 participants (23%) who completed the follow-up had harmed themselves more than five times in the year before outpatient department presentation on

FASM. 5 participants (38%) had harmed themselves less than five times, while another 5 participants (38%) reported no self-harm episode (other than the one leading to an outpatient department presentation) in the year before the intervention. On the follow-up assessment, only one participant (8%) had harmed himself more than five times during follow-up period, whereas 3 participants

**Table 2.** Linear mixed-effect models of change with ATMAN in BSI and PHQ scores post-intervention and at 10-month follow-up

| Variable | Baseline | Post-therapy | 10-month follow-up | Baseline vs. post-therapy | p-Value | Baseline vs. 10-month follow-up | p-Value |
|---|---|---|---|---|---|---|---|
| **BSI** | Mean (95% CI) | Mean (95% CI) | Mean (95% CI) | Mean difference (95%CI) | | Mean difference (95%CI) | |
| Unadjusted | 22.7 (19.4, 26.0) | 8.6 (6.2, 11.1) | 5.6 (3.7, 7.5) | −14.1 (−17.2, −10.9) | <0.001 | −17.0 (−20.4, −13.5) | <0.001 |
| Model 1 – Adjusted for duration of intervention | 22.6 (19.4, 25.9) | 8.6 (6.1, 11) | 5.7 (3.7, 7.6) | −14.1 (17.2, 10.9) | <0.001 | −17.0 (−20.4, −13.5) | <0.001 |
| Model 2 – Adjusted for age, gender and duration of intervention | 22.6 (19.6, 25.6) | 8.6 (6.1, 11.1) | 5.6 (3.9, 7.4) | −14.1 (−17.2, −10.9) | <0.001 | −17.0 (−20.4, −13.6) | <0.001 |
| Model 3 – Adjusted for specialist vs. non-specialist provider, age, gender and duration of intervention | 22.5 (19.7, 25.4) | 8.5 (6.4, 10.6) | 5.8 (3.8, 7.9) | −14.1 (−17.2, −10.9) | <0.001 | −16.7 (−20.1, −13.4) | <0.001 |
| **PHQ** | Mean (95% CI) | Mean (95% CI) | Mean (95% CI) | Mean difference (95%CI) | | Mean difference (95%CI) | |
| Unadjusted | 15.5 (12.0, 19.0) | 5.9 (3.7, 8.0) | 5.0 (3.3, 6.6) | −9.6 (−12.8, −6.4) | <0.001 | −10.5 (−14.5, −6.6) | <0.001 |
| Model 1 – Adjusted for duration of intervention | 15.5 (12.1. 19.0) | 5.9 (3.7, 8.1) | 4.9 (3.2, 6.6) | −9.6 (−12.8, −6.4) | <0.001 | −10.6 (−14.6, −6.6) | <0.001 |
| Model 2 – Adjusted for age, gender and duration of intervention | 15.5 (12.5, 18.5) | 5.9 (3.5, 8.2) | 4.8 (2.9, 6.8) | −9.6 (−12.8, −6.4) | <0.001 | −10.7 (−14.8, −6.6) | <0.001 |
| Model 3 – Adjusted for specialist vs. non-specialist provider, age, gender and duration of intervention | 15.5 (12.5, 18.5) | 5.9 (3.5, 8.2) | 4.8 (2.8, 6.7) | −9.6 (−12.8, −6.4) | <0.001 | −10.7 (−14.8, −6.6) | <0.001 |

Abbreviations: BSI, Beck's Suicidal Ideation Scale; PHQ, Patient Health Questionnaire; CI, confidence interval.

**Table 3.** Linear mixed-effect models of change

| BSI | Unadjusted | | Model 1 | | Model 2 | | Model 3 | |
|---|---|---|---|---|---|---|---|---|
| | β (95%CI) | p-Value | β (95%CI) | p-Value | β (95%CI) | p-Value | β (95%CI) | p-Value |
| Time (ref-baseline) | | | | | | | | |
| Post-intervention | −14.1 (−17.2, −10.9) | <0.001 | −14.1 (−17.2, −10.9) | <0.001 | −14.1 (−17.2, −10.9) | <0.001 | −14.1 (−17.2, −10.9) | <0.001 |
| After 10 months | −17 (−20.5, −13.6) | <0.001 | −17 (−20.4, −13.5) | <0.001 | −16.9 (−20.3, −13.6) | <0.001 | −16.7 (−20.1, −13.4) | <0.001 |
| Duration of intervention (days) | | | −0.03 (−0.11, 0.05) | 0.469 | −0.02 (−0.08, 0.04) | 0.467 | 0.02 (−0.03, 0.07) | 0.479 |
| Age (years) | | | | | −0.43 (−0.87, 0.01) | 0.057 | −0.81 (−1.36, −0.25) | 0.004 |
| Gender male (ref-female) | | | | | 2.78 (−1.54, 7.1) | 0.207 | 5.45 (1.39, 9.51) | 0.008 |
| Specialist (ref-non-specialist | | | | | | | −4.91 (−9.77, −0.06) | 0.047 |

| PHQ 9 | Unadjusted | | Model 1 | | Model 2 | | Model 3 | |
|---|---|---|---|---|---|---|---|---|
| | β (95%CI) | p-Value | β (95%CI) | p-Value | β (95%CI) | p-Value | β (95%CI) | p-Value |
| Time (ref-baseline) | | | | | | | | |
| Post-intervention | −9.6 (−12.8, −6.4 | <0.001 | −9.6 (−12.8, −6.4) | <0.001 | −9.6 (−12.8, −6.4) | <0.001 | −9.6 (−12.8, −6.4) | <0.001 |
| After 10 months | −10.5 (−14.5, −6.6) | <0.001 | −10.6 (−14.6, −6.6) | <0.001 | −10.7 (−14.8, −6.6) | <0.001 | −10.7 (−14.8, −6.6) | <0.001 |
| Duration of intervention (days) | | | 0.01 (−0.06, 0.09) | 0.722 | 0.02 (−0.03, 0.08) | 0.452 | 0.02 (−0.05, 0.08) | 0.629 |
| Age (years) | | | | | −0.41 (−0.86, 0.05) | 0.084 | −0.35 (−1.05, 0.34) | 0.321 |
| Gender male (ref-female) | | | | | 2.83 (−0.49, 6.15) | 0.095 | 2.45 (−2.57, 7.46) | 0.339 |
| Specialist (ref-non-specialist | | | | | | | 0.68 (−4.99, 6.34) | 0.815 |

(23%) harmed themselves once. There was no self-harm reported in 9 participants (69%).

### Qualitative interview findings

Qualitative exit interviews were completed with 17 young people at the time of intervention completion; exit interview was not conducted with one client who moved out of home soon after the intervention and could not be contacted for the interview. 13 young people participated in interviews at 10-month follow-up. Qualitative outcome measures complemented the quantitative assessments by allowing us to understand which intervention components were most valued by the participants and the ways the strategies learned during the intervention were used by the participants during the follow-up period. Themes from exit interviews included the following: (i) the importance of collaborative intervention planning and individual choices; (ii) value of flexibility in treatment schedule (with a possibility of reducing the number of sessions, involving family as per the choice of the young person); (iii) increased understanding of self-harm a deterrent to self-harm; (iv) the importance of practicing strategies over time; (v) the use of strategies beyond self-harm and (vi) the need to address stigma associated with self-harm.

### Intervention planning a collaborative process

11 young people felt heard and their choices were respected during the intervention process. Engagement with the therapist was deemed by the young people to be very important to their intervention experience. Young people felt that the intervention was tailored for them and they had an active role in the process.

"Someone actually listened to me" (F,19).

"Intervention involved me in it" (F,17).

### Value of flexibility in treatment schedule

Nine young people suggested that the flexible treatment schedule and the option of including family in ways they wanted to, were very useful for them. It allowed them to participate in intervention according to their needs.,

The choice to decide how long the participants wanted to continue intervention was particularly useful for a couple of young people who used limited treatment sessions. "I was well after two sessions. Knowing that I could decide when to finish treatment meant I was not forcing myself to attend unnecessary sessions. I practiced strategies to control my anger outbursts, which I learnt during those two sessions." (M, 24)

Involving family members was particularly valuable for five young people. "My mother and aunt could understand why I was harming myself and how to help me when I experienced these thoughts. I could reach out to them for support. This helped me a lot." (F,15)

### Increased understanding of self-harm a deterrent

An important theme in the interviews of nine young people was how improved understanding about the reasons for self-harm acted as a deterrent. Knowing their reasons for self-harm and identifying situations when it was likely to happen helped young people to reduce it. They would recognise the early warning signs and remove themselves from the situation to prevent self-harm. Two participants mentioned that improved understanding about self-harm by their families allowed the family members to support them better.

Using strategies beyond self-harm. Seven participants suggested they used strategies from the ATMAN intervention to deal with daily life difficult situations and distress.

"When we made the relationship map, it helped me to identify who was important to me and who was not important to me. When unimportant people caused me trouble during the last few months, I just remembered that map and thought how these people did not really matter to me." (F,15)

### Understanding strategies over time.

Five participants suggested that they were able to understand which strategies were most helpful for them and the situations in which these strategies worked the best, over a period of time.

"I was able to control my anger and sadness using the self-soothing strategies we worked on. It was difficult initially but it got better over time." (F,17)

Three of them reported that while going through the intervention process, they felt nothing was working. However, they found the strategies helpful after using them a few times with much effect.

"Looks like intervention and medicines helped me survive that phase. I felt nothing was working then." I realised the value of it all a bit later." (M,20)

"Hard to understand what helped but something did. Between completing intervention and now, the situation was out of my control many times but I managed to survive it without self-harm". (F, 15)

### Need to address stigma

Four young people spoke about shame associated with self-harm being a deterrent to help-seeking and engagement with the therapist.

One young person suggested, "You don't have to make me feel bad about harming myself. I am already feeling that. Help me overcome it in a respectful manner" (F, 17). Young people recognised stigma (prejudice of the mental health providers towards those who self-harm) as a barrier to engagement that could later results in the breakdown of the therapeutic relationship.

## Discussion

ATMAN shows promising effects in reducing self-harm thoughts and behaviour post-intervention when delivered by both specialist and non-specialist providers. The positive effects of the intervention were maintained at the 10-month follow-up. Feeling heard, having a say in their own treatment schedule, and not feeling judged or stigmatised during the ATMAN treatment were the themes that emerged during the exit interviews of the participants. Furthermore, the perceived benefits of the intervention ranged from a better understanding of self-harm to knowing what to do when the thoughts of self-harm were experienced over a period of time.

We found that the BSI and PHQ-9 scores were lower at the 10-month follow-up as compared to post-intervention values (although not significantly different). This is in contrast to the findings of the systematic reviews that have shown diminishing therapeutic efficacy of psychological interventions in reducing self-harm thoughts over the follow-up period compared to the end of treatment (Witt et al. 2021). One reason could be the higher prevalence of underlying common mental health problems (in more than half of the participants) in the current case series, which could be more amenable to psychological treatments such as ATMAN (with elements of problem-solving), as compared to personality disorders requiring more intensive and long-term forms of therapy (Kothgassner et al. 2020).

There is a paucity of publicly funded mental health care in India, especially in rural communities, with two mental health workers and 0·3 psychiatrists per 100,000 population (Sagar et al. 2020). Young people presenting to mental health services deal with many challenges. A lack of services specifically designed to meet the needs of young people makes it difficult for them to get the desired help. With an extremely limited availability of psychological interventions, psychopharmacology is a standard treatment for most mental health problems, including self-harm, even in children and adolescents (Rathod et al. 2017). In addition, young people with self-harm who do receive psychological interventions report their experiences with any such interventions to be negative (Aggarwal et al. 2020).

In this context, ATMAN, with its brief and flexible (five to eight sessions) schedule and optional modules that can be chosen by the counsellor and the young person together, offers an advantage over other more intensive therapeutic interventions for self-harm. Although not cost-effective, longer-term treatments, such as DBT-A, have shown better efficacy in reducing self-harm in young people as compared to shorter-term treatments (NICE 2022; Witt et al. 2021). During our case series, we received feedback from many participants wanting an additional session or expressing a desire to repeat the treatment schedule. Having its own advantages (such as a greater use in resource-limited settings), the brief schedule of ATMAN intervention restricts the period of contact between the counsellor and the young person. However, ATMAN has a provision for the counsellor to use additional sessions if needed. The counsellor can schedule an additional session to check in on the young person after 6–8 months (earlier, if deemed appropriate) following the treatment intervention completion and schedule the follow-up session during the last session of the intervention as part of the discharge. This may make the young person feel more supported and reach out for help when needed. However, ATMAN is not appropriate in situations where the young person is experiencing significant suicidal ideations with an imminent plan to end their life requiring an inpatient admission.

In our study, an improved understanding of self-harm by the young person and their family, and how best to use strategies, helped young people deal with it adequately. Recovery from self-harm is a subjective process aided by meaning-making of self-harm as well as having a sense of purpose and goals in the context of distress (Lewis and Hasking 2021). Furthermore, it is possible that the recovery trajectory of self-harm in young people in LMICs differs from young people in HICs, with a course that is more dependent on social factors as compared to recovery from mental health issues (Aggarwal et al. 2017; Knipe et al. 2019). An additional consideration would be the help-seeking attitudes and stigma related to self-harm in young people in LMICs (Aggarwal et al. 2021c; Bruffaerts et al. 2011). Indigenously developed or adapted psychological treatments such as ATMAN may align better with the needs of young people in LMICs by taking into account the contextually informed explanatory models (Aggarwal et al. 2020). Any cultural adaptation of ATMAN would require (i) taking into account the cultural concept of self-harm and the distress it causes; (ii) ensuring acceptability of the treatment components and (iii) incorporating culturally appropriate case vignettes for adequate treatment delivery (Heim and Kohrt 2019). Additionally, the scalability of the intervention can allow the integration of ATMAN at various levels of healthcare to manage and prevent the recurrence of self-harm in young people.

Young people participating in this case series continued to take the medications prescribed by the treating team. Since this was a

pragmatic study, we wanted to assess the effects of the ATMAN intervention in situations close to real life. It is possible that the improvement experienced by some young people, especially in reducing their depressive symptoms, was due to the medications they were taking. However, limited benefit of medications in reducing self-harm and suicidal thoughts is one of the strongest arguments in favour of using psychological interventions, as recommended by various international guidelines (N.I.C.E. 2022). In addition, our sample size was small due to the recruitment challenges we faced during the COVID-19 pandemic. Yet, our findings are important due to the limited studies that have assessed the effectiveness of available psychological interventions in reducing self-harm recurrence in young people in LMICs (Aggarwal et al. 2021b). Fewer still have explored adaptations of existing or development of newer interventions in LMICs. Such studies are important considering that self-harm is being increasingly recognised as a unique clinical presentation with multiple underlying causes, including mental disorders and social factors. Available evidence suggests a less robust association between self-harm and mental health problems in LMICs as compared to HICs and a stronger association with social causes (Aggarwal et al. 2017; Knipe et al. 2019). Thus, it becomes important to adapt and evaluate psychological interventions for self-harm in LMICs rather than generalising the evidence available from HICs to these settings.

## Conclusion

Improving support for young people who self-harm in LMICs is a priority for suicide prevention in this age group globally. We have shown that ATMAN is a contextually adapted intervention which when delivered by specialist and non-specialist counsellors alike, helps in reducing self-harm thoughts and behaviours. The positive effects were maintained at the 10-month follow-up. Although requiring evaluation in a randomised controlled trial, our results are significant in the absence of scalable interventions available for youth self-harm that can be used in LMICs to reduce the burden of suicide in this age group.

**Open peer review.** To view the open peer review materials for this article, please visit http://doi.org/10.1017/gmh.2025.26.

**Supplementary material.** The supplementary material for this article can be found at http://doi.org/10.1017/gmh.2025.26.

**Data availability statement.** The authors confirm that the data supporting the findings of this study are available within the article and its supplementary materials.

**Author contribution.** S Aggarwal was involved in conceptualising, designing and executing the study, analysis and drafting the manuscript. G Patton was involved in conceptualising and designing the study. M Berk and V Patel were involved in guiding the study design, analysis and drafting the manuscript. N Shah helped in recruitment for the study and guided the study design. A Shah was involved in data collection and analysis. D Kondal was involved in data analysis and interpretation of the results.

**Financial support.** This work was supported by the Wellcome Trust-India Alliance Research Fellowship awarded to the first author (grant number – IA/CPHE/16/1/502664). M Berk is supported by an NHMRC Senior Principal Research Fellowship (1156072).

**Competing interest.** The authors declare no conflict of interest.

**Ethics and consent.** The authors assert that all procedures contributing to this work comply with the ethical standards of the relevant national and institutional committees on human experimentation and with the Helsinki Declaration of 1975, as revised in 2008.

The study was reviewed and approved by three ethics committees: Institutional Ethics Committee Lokmanya Tilak Municipal Medical College & General Hospital, Mumbai-IEC/787121, Institutional Ethics Committee Public Health Foundation of India, Gurgaon, Haryana IEC-366.1/17 and Deakin University Human Research Ethics Committee, Melbourne Burwood Campus, Australia x-ref 2020-230.

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
