## [Reviewer Report]

Overall comments: The researchers aimed to examine the effectiveness of ATMAN in reducing self-harm thoughts among the youth population in India. They employed a mixed-method case study design to address this objective. Their findings suggest that ATMAN was significantly effective, regardless of whether it was delivered by specialist mental health professionals or non-specialists.

However, upon review, I found that the manuscript does not clearly articulate the problem statement. Additionally, the researchers did not adhere to ethical standards, notably violating the exclusion criteria during participant recruitment. They also provided therapeutic support by non-specialist professionals to participants with histories of self-harm and other severe psychological issues. Furthermore, the data analysis was superficial.

In conclusion, while this manuscript has potential, I cannot recommend it for publication in its current form.

Impact statement: In line 28, researchers claim, “an absence of definitive evidence for the effectiveness of available interventions in reducing self-harm in children and adolescents”. There are several articles to prove that DBT is effective in reducing self-harm for children and adolescents.

Furthermore, it will be better to rewrite the impact statement. For instance, there is no need to explain what ATMAN intervention is. The authors have focused on the background information and description of ATMAN instead of the impact of your present article.

Background:

I am very confused about the ATMAN because of its modules. All intervention techniques are borrowed from other evidence based psychological therapies. For example, problem solving is a technique of CBT, and emotion regulation is a technique of DBT which are included in the module of ATMAN. It will be better if the authors add a paragraph explaining the novelty of ATMAN intervention.

In the problem statement, it is not clear why the researchers provided ATMAN intervention by both specialist and non-specialist providers (103). My concern is about the ethical consideration in providing services by non-specialists when the cases had self-harm histories. Additionally, authors mentioned that non-specialists had no experience of delivery psychotherapy, and they had no adequate educational knowledge (e.g., master’s degree or MPhil degree) (150). Dealing self-harm patients by non-specialist professionals make them more vulnerable and it is considered as violation of safe clinical practice. In 246, authors mentioned that there was a significant threat to the safety of the participants or others around them. Hence, was it ethical to provide psychological interventions by non-specialists? Could you please provide the justification for it?

Method: In 109, there is a typo. Please write “from” an out-patient instead of “form”. The population of Mumbai city will be more because you have cited the data from 2011. In the recruitment process, it is mentioned that “youth with conditions that could interfere in their capacity to participate in treatment” were excluded. But Figure 1 (609-626) indicated that 2 participants dropped out because of having significant cognitive deficits for 1 participant and because of having severe delusional disorder. Hence, the recruitment contradicted the exclusion criteria.

Furthermore, in the methodology section, there is no depiction of research design the researchers used in this study. Why did they employ mixed method case series design? Please explain it with citations.

There is redundancy in explaining ATMAN. For example, the same information has been provided in 17-22, 91-96, 124.

For outcome measures, it is not clear why researchers did not set a specific time for measuring outcome for each participant. For example, measuring post outcome from one to two months from the start of the intervention is not feasible to compare the outcome for each participant.

For the qualitative part of the study, authors stated that they developed an interview guide, but they did not provide the questionnaires. Was it structured or semi-structured interview guides? At the first glance, the purpose of the qualitative part seems to explore the difficulties in understanding the modules of ATMAN intervention. It is not clear how the qualitative part relates to the quantitative part where they conducted this study to examine the effectiveness of ATMAN intervention. Could you please provide your justification? It seems that quantitative and qualitative methods are two separate methods here.

For the quantitative part of the study, authors did not mention anything how they controlled extraneous variables to examine the effectiveness of ATMAN intervention.

Data analysis: Researchers employed phenomenological thematic analysis, though the interviews, and depth and nature of the data did not match the requirements for phenomenological data analysis. Please provide your justification why it is phenomenological thematic analysis. The qualitative data was analyzed by two researchers. Did they check the intercoder agreement or use any data analysis tools?

Results and discussion

Sample should be discussed in the methodology section.

For outcome section-the organization of the findings is not clear. It is not clear if specialist’s participants or non-specialist’s participants conducted self-harm.

It is not clear whether all participants receive the same treatment modules or not. If they did not receive the same treatment modules, how the effectiveness was determined especially the comparison between specialist and non-specialist service providers?

Participants took antidepressants and antipsychotic medications. How the researchers were able to exclude the impact of the medication to measure the effectiveness of ATMAN?

---

## [Reviewer Report]

The manuscript presents the ATMAN intervention, designed for youths that have self-harmed. The research is interesting and helps to have more information for intervention with patients who practice self-harm. I applaud the effort of the authors to design, implement, test and publish this work. I am always glad to find well done research from LMICs.Very ethical to consider risk management on this population. Following, some comments to improve the content.

Mayor recommendations

Background

- I recommend to expand the background on other similar psychological intervention projects that have been applied to equitable contexts and their results.

- I miss a clear objective of the intervention (not of the manuscript). Is it reducing self-harm recidiving? Reducing suicide ideation?

- I miss information regarding the available mental health services in India for the general population, pros and cons.

Methods

- Authors did not mention anything about informed participant/parental consent and/or assent. What is the legal age to sign an informed consent in India?

- “Proficiency in written and spoken Hindi or English”. So not all participants used the same language?, Both providers were bilingual? Would it be a variable to be considered as intervinient? What percentage of participants took ALTMAN in hindi/english?

-Include an explanation or description of why these psychological measurement instruments were chosen and their effectiveness in measuring the variables in your population (translation and validation).

-Deep into the description of the interviews and subsequent applications on the follow-up after 10 months to the participants.

- What was the criteria to follow-up at 10 months, when most research does at 6 and 12 months?

- To my understanding not all the participants had the very same intervention (optional sessions), wouldn´t it be a intervenient variable to consider?

- “providers used video-recorded sections of their own therapy sessions and role-playing to discuss various elements of each session. In addition, the video recordings of therapy sessions delivered by the counsellors were used to give feedback by the specialist providers during regular supervision”. Who were recorded and why? The patients and/or the providers? How do patients agree to be filmed and later exposed?, were these recordings of individual or group sessions? I am concerned about confidentiality of content, particularly when dealing with youths.

- what type of mental health services are available for the participants on regular basis?

I miss what I consider relevant information regarding ATMAN

- Detail about the sessions (individual or group? location? for how long? resource provision?, schedule?)

- When designing the intervention, what theoretical (e.g. cbt?) and methodological (e.g. see https://doi.org/10.1136/jech-2015-205952) frameworks did authors follow?

- What was the role of users in the design of ATMAN?

- Most important, provide enough information for the readers to be able to reproduce as much as possible the study if interested.

Results

- Add the cut-off points that were taken as indicators that the intervention had positive effects on the population.

- I would like to see a graphic with three horizontal lines, one representing the individual BPI scores before the intervention and the other two with post ATMAN- and follow up- BPI scores. Same for PHQ-9 scores.

- It would enrich the text reporting if there where any cases of users not reporting the expected outcome individually (same scores or scores on the negative direction after ATMAN.

Discussion

- The background states that the objective of the research was to “generate the preliminary effects of ATMAN intervention in reducing self-harm thoughts when delivered by specialist and non-specialist providers; to assess if the effects are sustained at 10-months follow-up; and to assess therapy experience at each step”. The one scale for the above mentioned dependent variable (self-harm thoughts) was the Columbia- Suicide severity rating scale (C-SSRS), which did not show significant changes from base line to end of intervention, neither at follow-up. Considering the stated objective, this would be the most relevant result to discuss.

- Provide recommendations for upcoming replications of ATMAN

- Comment in what conditions ATMAN would be not advisable and why

- Do authors consider feasible a further follow-up? Why or why not?

Minor recommendations

Page 1. “Self-harm which is defined . . .). Cite source.

Page 1 and others. First time a acronym is mentioned authors must give the complete noun.

Page 1, line 13. “developed and tested in high-income countries (HICs) without being adapted”. What sort of adaptation do authors considered indispensable? Be specific.

Page 1, line 17 “ATMAN is one of the very few psychological interventions”. So there are others, apart from ATMAN. Those shall be listed and commented.

Page 1. Line 18. “developed with inputs from expert service providers as well as service users . . .”. This sounds interesting. If providers and users all together worked to develop and test ATMAN, it would be worth it to have all the process published.

Page 1. Line 23. “. . the current psychological treatments available for youth self-harm . . .” Which are these available treatments?

Page 1. Line 31. “. . . can help us develop newer more effective interventions”. Delete “newer”.

Page 3. Line 77. OrganiZation, not OrganiSation

Page 3. Line 79 What does NICE stand for?

Page 3. Line 97. Clarify, ¿anyone regardless of their skills? Would the provider need training for ATMAN?, what would the training be like (self-learning?, how long?, minimum passing grade?)

Page 5. Line 146. It would be helpful for readers from different countries to explain how health providers are qualified in India.

---

## [Reviewer Report]

Review of the Article: “Effectiveness of ATMAN Psychological Intervention in Reducing Self-Harm in Young People in India: A Mixed Method Case Series”

**General Suggestions:**

1. **Terminology Consistency**: The terms “therapy” and “intervention” should not be used interchangeably in this study. The term “therapy” implies a broad spectrum of approaches, while “intervention” here refers to a specific modular program. Consistency in terminology will help maintain clarity.

**Specific Comments:**

- **Line 10**: The phrase “clearest risk factor” is not appropriate. I suggest replacing it with “identified risk factor” or stating that “research has shown evidence supporting its relation to...”. This adjustment should also be applied to Line 72.

- **Line 14**: Clarify what is meant by “where most young people live.” Consider specifying the context or demographic details.

- **Line 28**: The phrase “in a series of young people” is ambiguous. It is unclear whether this implies a rotational basis or another structure. Please rephrase for clarity.

- **Line 39**: In line 93, it is noted that ATMAN is a Sanskrit word. It would be more appropriate to italicize it rather than capitalize it.

- **Line 42**: The phrase “youth with self-harm” needs clarification. Does this refer to youth currently engaged in self-harm, those with a history of self-harm, or those with suicidal intent? Precise language is needed to define the target population.

- **Line 43**: The term “completed therapy” implies a broad process. Clarify whether this means the participants completed all sessions planned in the protocol or if there is another definition.

- **Line 45**: Adjust the reporting of statistics, such as the mean difference and confidence intervals (e.g., 14.1 (17.2, 10.9)), to adhere to APA style guidelines. This correction should be applied to all statistical reporting throughout the paper.

- **Lines 72-73**: As the focus is on youth, it would be beneficial to report statistics specific to youth rather than the general population.

- **Line 77**: Please check the spelling of references.

- **Lines 97-99**: The specifics of the modifications made to the intervention should be clearly described. This information is more suitable for the Methods section rather than the Introduction. The implications of these modifications should then be discussed in the Discussion section.

- **Line 109**: Spell out “November” and use complete words. Consider revising to “January 2021 to November 2022.”

- **Line 111**: This line should not be in the Methods section as it is background information. A specific paragraph discussing Mumbai, its population, and studies related to self-harm and suicide in youth would be more appropriate. Remove this from the Methods section.

- **Line 141**: Correct the spelling of “counselor.”

**Intervention Section Suggestions:**

Provide additional details about the intervention:

- Was the intervention delivered online or in person?

- Was it a group-based therapy or individual sessions?

- What was the duration of the intervention (weeks/months)?

- How frequently did participants meet with the therapist each week?

- What was the duration of each session?

- How many sessions were required for a participant to be considered compliant with the treatment?

- How was fidelity to the intervention measured?

- Did the therapists receive supervision?

- Why did some participants attend fewer than five sessions? Was this accounted for in the analysis?

**Outcome Scores Section:**

Clarify the following points:

- Was the structured assessment designed by the researchers? If so, provide details.

- Report the reliability of the Brief Symptom Inventory (BSI) for this study, not just past studies.

- When stating the BSI is “widely used,” provide relevant studies, especially considering that this is an intervention culturally adapted for Indian youth.

- For the Columbia-Suicide Severity Rating Scale (C-SSRS), what reliability score is considered acceptable? Provide specific numbers.

- For the Patient Health Questionnaire-9 (PHQ-9), clarify whether it was administered after each session or at the end of the intervention. Include references that validate its use with Indian adolescents.

- For the Functional Assessment of Self-Mutilation (FASM), the statement “administered during the end of therapy assessment; the timeframe for FASM questions was from the start of therapy till the end of therapy” is unclear. Clarify and provide reliability scores (e.g., Cronbach’s alpha).

- Regarding the Global Assessment of Functioning Scale (GAS), clarify whether it is widely used among Indian youth and report its reliability. Specify if the global or children’s version was used.

**Sample Section:**

Include details on any exclusion criteria for adolescents.

**Results Section:**

The quantitative analysis needs further development:

- Complete the tables by reporting beta coefficients, standard errors, and model parameters. Describe these elements in the text as well.

- Consider the effect of the therapist by including an interaction effect to determine if scores differ based on these differences. This should be described in the text and represented in the tables.

- Address the handling of missing data. Did the authors assume data were missing completely at random and analyze only complete cases? If so, consider conducting an analysis with intent to treat.

- Provide a clear explanation of group assignment. Was participant distribution random? Mention the use of a PRISMA diagram to help readers understand the trial flow.

**Discussion Section:**

- Provide a clear rationale for using specialists versus non-specialists in the intervention. Discuss the implications of this comparison for the discipline.

- The intervention aims to be adaptable to Indian youth, but there is no mention of cultural values integrated into each session. This should be highlighted to distinguish this intervention from others.

- A significant limitation that needs to be discussed is that participants were receiving therapy and medication in an outpatient setting. There is no mention of the type of therapy or medication, and how these factors might affect the study results. The absence of a control group makes it incorrect to assume that the intervention alone led to improvement; this should be addressed as a limitation.

---

## [Editor Report]

The reviewers consider that the article requires a major revision, correcting some parts, expanding the information and other details Because the ATMAN intervention could be an interesting contribution, the comments of the three reviewers are described in detail, with the aim that the authors review them and send a new version.

Reviewer 1: Overall comments: The researchers aimed to examine the effectiveness of ATMAN in reducing self-harm thoughts among the youth population in India. They employed a mixed-method case study design to address this objective. Their findings suggest that ATMAN was significantly effective, regardless of whether it was delivered by specialist mental health professionals or non-specialists.

However, upon review, I found that the manuscript does not clearly articulate the problem statement. Additionally, the researchers did not adhere to ethical standards, notably violating the exclusion criteria during participant recruitment. They also provided therapeutic support by non-specialist professionals to participants with histories of self-harm and other severe psychological issues. Furthermore, the data analysis was superficial.

In conclusion, while this manuscript has potential, I cannot recommend it for publication in its current form.

Impact statement: In line 28, researchers claim, “an absence of definitive evidence for the effectiveness of available interventions in reducing self-harm in children and adolescents”. There are several articles to prove that DBT is effective in reducing self-harm for children and adolescents.

Furthermore, it will be better to rewrite the impact statement. For instance, there is no need to explain what ATMAN intervention is. The authors have focused on the background information and description of ATMAN instead of the impact of your present article.

Background:

I am very confused about the ATMAN because of its modules. All intervention techniques are borrowed from other evidence based psychological therapies. For example, problem solving is a technique of CBT, and emotion regulation is a technique of DBT which are included in the module of ATMAN. It will be better if the authors add a paragraph explaining the novelty of ATMAN intervention.

In the problem statement, it is not clear why the researchers provided ATMAN intervention by both specialist and non-specialist providers (103). My concern is about the ethical consideration in providing services by non-specialists when the cases had self-harm histories. Additionally, authors mentioned that non-specialists had no experience of delivery psychotherapy, and they had no adequate educational knowledge (e.g., master’s degree or MPhil degree) (150). Dealing self-harm patients by non-specialist professionals make them more vulnerable and it is considered as violation of safe clinical practice. In 246, authors mentioned that there was a significant threat to the safety of the participants or others around them. Hence, was it ethical to provide psychological interventions by non-specialists? Could you please provide the justification for it?

Method: In 109, there is a typo. Please write “from” an out-patient instead of “form”. The population of Mumbai city will be more because you have cited the data from 2011. In the recruitment process, it is mentioned that “youth with conditions that could interfere in their capacity to participate in treatment” were excluded. But Figure 1 (609-626) indicated that 2 participants dropped out because of having significant cognitive deficits for 1 participant and because of having severe delusional disorder. Hence, the recruitment contradicted the exclusion criteria.

Furthermore, in the methodology section, there is no depiction of research design the researchers used in this study. Why did they employ mixed method case series design? Please explain it with citations.

There is redundancy in explaining ATMAN. For example, the same information has been provided in 17-22, 91-96, 124.

For outcome measures, it is not clear why researchers did not set a specific time for measuring outcome for each participant. For example, measuring post outcome from one to two months from the start of the intervention is not feasible to compare the outcome for each participant.

For the qualitative part of the study, authors stated that they developed an interview guide, but they did not provide the questionnaires. Was it structured or semi-structured interview guides? At the first glance, the purpose of the qualitative part seems to explore the difficulties in understanding the modules of ATMAN intervention. It is not clear how the qualitative part relates to the quantitative part where they conducted this study to examine the effectiveness of ATMAN intervention. Could you please provide your justification? It seems that quantitative and qualitative methods are two separate methods here.

For the quantitative part of the study, authors did not mention anything how they controlled extraneous variables to examine the effectiveness of ATMAN intervention.

Data analysis: Researchers employed phenomenological thematic analysis, though the interviews, and depth and nature of the data did not match the requirements for phenomenological data analysis. Please provide your justification why it is phenomenological thematic analysis. The qualitative data was analyzed by two researchers. Did they check the intercoder agreement or use any data analysis tools?

Results and discussion

Sample should be discussed in the methodology section.

For outcome section-the organization of the findings is not clear. It is not clear if specialist’s participants or non-specialist’s participants conducted self-harm.

It is not clear whether all participants receive the same treatment modules or not. If they did not receive the same treatment modules, how the effectiveness was determined especially the comparison between specialist and non-specialist service providers?

Participants took antidepressants and antipsychotic medications. How the researchers were able to exclude the impact of the medication to measure the effectiveness of ATMAN?

Reviewer 2:The manuscript presents the ATMAN intervention, designed for youths that have self-harmed. The research is interesting and helps to have more information for intervention with patients who practice self-harm. I applaud the effort of the authors to design, implement, test and publish this work. I am always glad to find well done research from LMICs.Very ethical to consider risk management on this population. Following, some comments to improve the content.

Mayor recommendations

Background

- I recommend to expand the background on other similar psychological intervention projects that have been applied to equitable contexts and their results.

- I miss a clear objective of the intervention (not of the manuscript). Is it reducing self-harm recidiving? Reducing suicide ideation?

- I miss information regarding the available mental health services in India for the general population, pros and cons.

Methods

- Authors did not mention anything about informed participant/parental consent and/or assent. What is the legal age to sign an informed consent in India?

- “Proficiency in written and spoken Hindi or English”. So not all participants used the same language?, Both providers were bilingual? Would it be a variable to be considered as intervinient? What percentage of participants took ALTMAN in hindi/english?

-Include an explanation or description of why these psychological measurement instruments were chosen and their effectiveness in measuring the variables in your population (translation and validation).

-Deep into the description of the interviews and subsequent applications on the follow-up after 10 months to the participants.

- What was the criteria to follow-up at 10 months, when most research does at 6 and 12 months?

- To my understanding not all the participants had the very same intervention (optional sessions), wouldn´t it be a intervenient variable to consider?

- “providers used video-recorded sections of their own therapy sessions and role-playing to discuss various elements of each session. In addition, the video recordings of therapy sessions delivered by the counsellors were used to give feedback by the specialist providers during regular supervision”. Who were recorded and why? The patients and/or the providers? How do patients agree to be filmed and later exposed?, were these recordings of individual or group sessions? I am concerned about confidentiality of content, particularly when dealing with youths.

- what type of mental health services are available for the participants on regular basis?

I miss what I consider relevant information regarding ATMAN

- Detail about the sessions (individual or group? location? for how long? resource provision?, schedule?)

- When designing the intervention, what theoretical (e.g. cbt?) and methodological (e.g. see https://doi.org/10.1136/jech-2015-205952) frameworks did authors follow?

- What was the role of users in the design of ATMAN?

- Most important, provide enough information for the readers to be able to reproduce as much as possible the study if interested.

Results

- Add the cut-off points that were taken as indicators that the intervention had positive effects on the population.

- I would like to see a graphic with three horizontal lines, one representing the individual BPI scores before the intervention and the other two with post ATMAN- and follow up- BPI scores. Same for PHQ-9 scores.

- It would enrich the text reporting if there where any cases of users not reporting the expected outcome individually (same scores or scores on the negative direction after ATMAN.

Discussion

- The background states that the objective of the research was to “generate the preliminary effects of ATMAN intervention in reducing self-harm thoughts when delivered by specialist and non-specialist providers; to assess if the effects are sustained at 10-months follow-up; and to assess therapy experience at each step”. The one scale for the above mentioned dependent variable (self-harm thoughts) was the Columbia- Suicide severity rating scale (C-SSRS), which did not show significant changes from base line to end of intervention, neither at follow-up. Considering the stated objective, this would be the most relevant result to discuss.

- Provide recommendations for upcoming replications of ATMAN

- Comment in what conditions ATMAN would be not advisable and why

- Do authors consider feasible a further follow-up? Why or why not?

Minor recommendations

Page 1. “Self-harm which is defined . . .). Cite source.

Page 1 and others. First time a acronym is mentioned authors must give the complete noun.

Page 1, line 13. “developed and tested in high-income countries (HICs) without being adapted”. What sort of adaptation do authors considered indispensable? Be specific.

Page 1, line 17 “ATMAN is one of the very few psychological interventions”. So there are others, apart from ATMAN. Those shall be listed and commented.

Page 1. Line 18. “developed with inputs from expert service providers as well as service users . . .”. This sounds interesting. If providers and users all together worked to develop and test ATMAN, it would be worth it to have all the process published.

Page 1. Line 23. “. . the current psychological treatments available for youth self-harm . . .” Which are these available treatments?

Page 1. Line 31. “. . . can help us develop newer more effective interventions”. Delete “newer”.

Page 3. Line 77. OrganiZation, not OrganiSation

Page 3. Line 79 What does NICE stand for?

Page 3. Line 97. Clarify, ¿anyone regardless of their skills? Would the provider need training for ATMAN?, what would the training be like (self-learning?, how long?, minimum passing grade?)

Page 5. Line 146. It would be helpful for readers from different countries to explain how health providers are qualified in India. 

Reviewer 3: **General Suggestions:**

1. **Terminology Consistency**: The terms “therapy” and “intervention” should not be used interchangeably in this study. The term “therapy” implies a broad spectrum of approaches, while “intervention” here refers to a specific modular program. Consistency in terminology will help maintain clarity.

**Specific Comments:**

- **Line 10**: The phrase “clearest risk factor” is not appropriate. I suggest replacing it with “identified risk factor” or stating that “research has shown evidence supporting its relation to...”. This adjustment should also be applied to Line 72.

- **Line 14**: Clarify what is meant by “where most young people live.” Consider specifying the context or demographic details.

- **Line 28**: The phrase “in a series of young people” is ambiguous. It is unclear whether this implies a rotational basis or another structure. Please rephrase for clarity.

- **Line 39**: In line 93, it is noted that ATMAN is a Sanskrit word. It would be more appropriate to italicize it rather than capitalize it.

- **Line 42**: The phrase “youth with self-harm” needs clarification. Does this refer to youth currently engaged in self-harm, those with a history of self-harm, or those with suicidal intent? Precise language is needed to define the target population.

- **Line 43**: The term “completed therapy” implies a broad process. Clarify whether this means the participants completed all sessions planned in the protocol or if there is another definition.

- **Line 45**: Adjust the reporting of statistics, such as the mean difference and confidence intervals (e.g., 14.1 (17.2, 10.9)), to adhere to APA style guidelines. This correction should be applied to all statistical reporting throughout the paper.

- **Lines 72-73**: As the focus is on youth, it would be beneficial to report statistics specific to youth rather than the general population.

- **Line 77**: Please check the spelling of references.

- **Lines 97-99**: The specifics of the modifications made to the intervention should be clearly described. This information is more suitable for the Methods section rather than the Introduction. The implications of these modifications should then be discussed in the Discussion section.

- **Line 109**: Spell out “November” and use complete words. Consider revising to “January 2021 to November 2022.”

- **Line 111**: This line should not be in the Methods section as it is background information. A specific paragraph discussing Mumbai, its population, and studies related to self-harm and suicide in youth would be more appropriate. Remove this from the Methods section.

- **Line 141**: Correct the spelling of “counselor.”

**Intervention Section Suggestions:**

Provide additional details about the intervention:

- Was the intervention delivered online or in person?

- Was it a group-based therapy or individual sessions?

- What was the duration of the intervention (weeks/months)?

- How frequently did participants meet with the therapist each week?

- What was the duration of each session?

- How many sessions were required for a participant to be considered compliant with the treatment?

- How was fidelity to the intervention measured?

- Did the therapists receive supervision?

- Why did some participants attend fewer than five sessions? Was this accounted for in the analysis?

**Outcome Scores Section:**

Clarify the following points:

- Was the structured assessment designed by the researchers? If so, provide details.

- Report the reliability of the Brief Symptom Inventory (BSI) for this study, not just past studies.

- When stating the BSI is “widely used,” provide relevant studies, especially considering that this is an intervention culturally adapted for Indian youth.

- For the Columbia-Suicide Severity Rating Scale (C-SSRS), what reliability score is considered acceptable? Provide specific numbers.

- For the Patient Health Questionnaire-9 (PHQ-9), clarify whether it was administered after each session or at the end of the intervention. Include references that validate its use with Indian adolescents.

- For the Functional Assessment of Self-Mutilation (FASM), the statement “administered during the end of therapy assessment; the timeframe for FASM questions was from the start of therapy till the end of therapy” is unclear. Clarify and provide reliability scores (e.g., Cronbach’s alpha).

- Regarding the Global Assessment of Functioning Scale (GAS), clarify whether it is widely used among Indian youth and report its reliability. Specify if the global or children’s version was used.

**Sample Section:**

Include details on any exclusion criteria for adolescents.

**Results Section:**

The quantitative analysis needs further development:

- Complete the tables by reporting beta coefficients, standard errors, and model parameters. Describe these elements in the text as well.

- Consider the effect of the therapist by including an interaction effect to determine if scores differ based on these differences. This should be described in the text and represented in the tables.

- Address the handling of missing data. Did the authors assume data were missing completely at random and analyze only complete cases? If so, consider conducting an analysis with intent to treat.

- Provide a clear explanation of group assignment. Was participant distribution random? Mention the use of a PRISMA diagram to help readers understand the trial flow.

**Discussion Section:**

- Provide a clear rationale for using specialists versus non-specialists in the intervention. Discuss the implications of this comparison for the discipline.

- The intervention aims to be adaptable to Indian youth, but there is no mention of cultural values integrated into each session. This should be highlighted to distinguish this intervention from others.

- A significant limitation that needs to be discussed is that participants were receiving therapy and medication in an outpatient setting. There is no mention of the type of therapy or medication, and how these factors might affect the study results. The absence of a control group makes it incorrect to assume that the intervention alone led to improvement; this should be addressed as a limitation.

---

## [Reviewer Report]

Overcall comments: Thank you to the authors for addressing all comments and for their intensive work to improve the quality of this manuscript. I am happy to accept this manuscript with minor corrections. I kindly request them to consider the following comments. Thank you.

Methods: Could you please include which mixed method case design (e.g., explanatory sequential, exploratory sequential, etc.) you have applied? I have understood your rationale of employing the mixed method case design, it will be better if you provide citation(s) against your rationale (for your assistance, you can may get help exploring the book by Creswell & Plano Clarke (2018); and Yin (2014).

Page 71 and 72 of 92 (line 378-385): “One participant had their outcome assessment scheduled earlier (at 5 weeks) because they were moving out of the study location”- please correct this statement, there is pronoun-antecedent agreement issue. For example, the noun is one participant, the pronoun can be his/her. Instead of they, it will be she/he.

One more suggestion for this section: the information from 378-381 are similar to information from 383-385. If I am not mistaken, please correct it, otherwise you can ignore this comment.

Thank you.

---

## [Editor Report]

Dear Dr Aggarwal.

The current version of your manuscript has been well evaluated by the reviewers. Congratulations!

We only ask that you consider three minor observations made by one of the reviewers, make corrections, and submit the latest version of the manuscript:

1) Could you please include which mixed method case design (e.g., explanatory sequential, exploratory sequential, etc.) you have applied? I have understood your rationale of employing the mixed method case design, it will be better if you provide citation(s) against your rationale (for your assistance, you can get help exploring the book by Creswell & Plano Clarke (2018); and Yin (2014).

2) Page 71 and 72 of 92 (line 378-385): “One participant had their outcome assessment scheduled earlier (at 5 weeks) because they were moving out of the study location”- please correct this statement, there is pronoun-antecedent agreement issue. For example, the noun is one participant, the pronoun can be his/her. Instead of them, it will be she/he.

3) One more suggestion for this section: the information from 378-381 are similar to information from 383-385. If I am not mistaken, please correct it, otherwise you can ignore this comment.

Thank you so much

---

## [Reviewer Report]

Thank you for your intensive work. Congratulations! Now, this manuscript is accepted for publication.

---

## [Editor Report]

The new version has been revised. It is a good work and contribution to this field. This version is accepted and congratulations to the authors.